# Low Dose Chest CT and Lung Ultrasound for the Diagnosis and Management of COVID-19

**DOI:** 10.3390/jcm10102196

**Published:** 2021-05-19

**Authors:** Julie Finance, Laurent Zieleskewicz, Paul Habert, Alexis Jacquier, Philippe Parola, Alain Boussuges, Fabienne Bregeon, Carole Eldin

**Affiliations:** 1IRD, APHM, MEPHI, IHU Méditerranée Infection, Aix Marseille University, 13005 Marseille, France; julie.finance@etu.univ-amu.fr (J.F.); Fabienne.BREGEON@ap-hm.fr (F.B.); 2Service des Explorations Fonctionnelles Respiratoires, APHM, 13005 Marseille, France; 3Department of Anaesthesiology and Intensive Care Medicine, Hôpital Nord, APHM, Aix Marseille Université, 13005 Marseille, France; laurent.zieleskewicz@ap-hm.fr; 4INRA, INSERM, Centre for Cardiovascular and Nutrition Research (C2VN), Aix Marseille Université, 13005 Marseille, France; alain.boussuges@ap-hm.fr; 5Service de Radiologie Cardio-Thoracique, Hôpital La Timone, APHM, 13005 Marseille, France; paul.habert@ap-hm.fr (P.H.); alexis.jacquier@ap-hm.fr (A.J.); 6LIIE, Aix Marseille University, 13005 Marseille, France; 7CNRS, CRMBM-CEMEREM (Centre de Résonance Magnétique Biologique et Médicale—Centre d’Exploration Métaboliques par Résonance Magnétique), APHM, Aix-Marseille University, UMR 7339, 13005 Marseille, France; 8IRD, APHM, SSA, VITROME, Aix Marseille University, 13005 Marseille, France; philippe.parola@univ-amu.fr; 9IHU-Méditerranée Infection, Aix Marseille University, 13005 Marseille, France

**Keywords:** lung ultrasound, low-dose CT, pneumonia, COVID-19, SARS-CoV-2

## Abstract

Background: The COVID-19 pandemic has provided an opportunity to use low- and non-radiating chest imaging techniques on a large scale in the context of an infectious disease, which has never been done before. Previously, low-dose techniques were rarely used for infectious diseases, despite the recognised danger of ionising radiation. Method: To evaluate the role of low-dose computed tomography (LDCT) and lung ultrasound (LUS) in managing COVID-19 pneumonia, we performed a review of the literature including our cases. Results: Chest LDCT is now performed routinely when diagnosing and assessing the severity of COVID-19, allowing patients to be rapidly triaged. The extent of lung involvement assessed by LDCT is accurate in terms of predicting poor clinical outcomes in COVID-19-infected patients. Infectious disease specialists are less familiar with LUS, but this technique is also of great interest for a rapid diagnosis of patients with COVID-19 and is effective at assessing patient prognosis. Conclusions: COVID-19 is currently accelerating the transition to low-dose and “no-dose” imaging techniques to explore infectious pneumonia and their long-term consequences.

## 1. Introduction

Pneumonia is the most frequent complication of COVID-19 infection and can lead to acute respiratory distress syndrome and the need for ventilatory support [1]. Before the COVID-19 pandemic, confirmation of pneumonia traditionally required the demonstration of a new-onset pulmonary infiltrate on chest X-rays (CXR) or on chest computed tomography (CT) in addition to consistent clinical symptoms and signs [2]. CXR is the most frequently used chest imaging technique but lacks sensitivity for the diagnosis of COVID-19 pneumonia [3]. CT is commonly considered as the gold standard for diagnosing pneumonia and the number of chest CTs performed has dramatically increased in recent years, together with total ionising radiation doses for patients [4]. However, ionising radiation damages tissues and alters DNA structure, which has been shown to increase the long-term cancer risk [5]. Low-dose (LD) and ultra-low-dose (ULD) CTs have been developed and applied to lung imaging in order to limit radiation exposure, but, usually, these are not routinely performed, except for lung cancer screening [6]. Moreover, lung ultrasounds (LUS), a non-radiating technique that can be quickly performed at the patient’s bedside, have also been shown to be accurate for the diagnosis of pneumonia [7]. LUS devices were initially reserved to intensivists but are now available in emergency departments (ED), respiratory care units, and in some infectious disease units, as is the case in the IHU Méditerrannée Infection. For viral pneumonia, however, its diagnostic value compared to CT remained unclear, until 2020. 

Currently, the need to rapidly evaluate patients with COVID-19 when they present with dyspnoea or other respiratory signs is an opportunity to use chest LDCT and LUS on a large scale. In March 2020, radiologists from our centre started to perform LDCT protocols with dedicated scanners for confirmed or suspected cases of COVID-19. Shortly after, chest LDCT was recommended as standard procedure by the British Thoracic Society for COVID-19 diagnosis [8]. This pandemic will, therefore, probably change the way we explore chest and diagnose infectious diseases. In this review, we aim to give an overview of the contribution of low- and non-radiating chest imaging techniques to the diagnosis of COVID-19 pneumonia and the role of these techniques in managing patients in this particular context.

## 2. Low-Dose Chest CT-Scan

### 2.1. Latest Developments before COVID-19

The goal of LDCT scanning is to maintain a good image resolution while reducing irradiation by optimising scanning parameters and by using iterative reconstruction algorithm [9]. Iterative reconstruction algorithms allow a dose reduction from 12% to 62% depending on solution used [10]. The first chest LDCT was performed in 1990 [11]. In this ground-breaking paper, Naidich et al. reported the same diagnostic image quality between LDCT and standard-dose CT scans in two patients with apparently normal lungs and in ten patients with a range of underlying parenchymal abnormalities (two of whom had pulmonary tuberculosis) [11]. A more widespread use of chest LDCT began in 2011 when this technique showed it could accurately reduce mortality as part of the lung cancer screening campaign in the USA [6]. ULDCT emerged in 2013, involving additional reduction of image quality that is again counterbalanced with innovative reconstruction techniques [12]. The definition of low- and ultra-low-dose chest CT does not include precise quantitative value or technical protocol. It is recognised that the radiation dose of chest LDCT should be half that of standard dose CT [13]. For lung cancer screening, the effective dose estimates were 1.6 and 2.4 mSv for one CT for men and women, respectively [6]. Chest ULDCT has a radiation dose equivalent to or lower than CXR (<1 mSv) [14]. In comparison, the effective dose for a conventional chest CT is estimated to be 3–4.8 mSv [15]. The performance of ULDCT and LDCT has been demonstrated for the diagnosis of pulmonary infections in prospective cohorts of immunocompromised patients [16], in comparison to standard-dose CT and microbial cultures. However, no guidelines existed about chest LD or ULDCT use during infectious diseases before the COVID-19 pandemic.

### 2.2. The Diagnosis of COVID-19 Pneumonia with LDCT

During the COVID-19 pandemic, there was a focus on using chest imaging to obtain an early diagnosis in patients with worsening respiratory status or risk factors of disease progression [17]. CXR showed poor sensitivity in mild or early COVID-19 infection [18]. The chest CT pattern [19] and the characteristic evolution of chest CT over time [20] were rapidly described as an important complementary technique for the diagnosis of COVID-19 pneumonia. Typical CT findings are represented by patchy ground-glass opacities, areas of consolidations, crazy-paving and bilateral multilobe consolidations in ICU patients [19]. LDCT had already been promoted as the first-line imaging technique by Chinese radiologists as early as March 2020 [21]. Radiologists began to routinely perform chest LDCTs to diagnose an infectious disease, allowing a rapid (5–10 min) triage of patients infected by COVID-19 with good radiation and efficacy ratios. 

In our centre, we performed 2065 LDCTs between February and May 2020 on 3737 COVID-19 patients, including 1449 (70.1%) that detected abnormalities [22]. LDCT scans were performed with the following parameters: detector collimation, 0.625 mm; field of view, 500 mm; matrix, 512 × 512; pitch, 1.375; gantry speed, 0.35 s; 120 KV; 45 mAs; and reconstructed slice thickness, 1.2 mm [23]. Interestingly, among 1108 patients who perceived themselves as non-dyspnoeic, 157 (14.2%) had an oxygen saturation <95% and LDCT revealed pneumonia in 139 of them [22]. In March 2020, the British Thoracic Society highlighted the need for a balance between minimising the radiation dose and ensuring high-quality diagnostic images and promoted the use of unenhanced chest LDCT as the standard-of-care for COVID-19 pneumonia imaging [8]. In July 2020, the international Fleishner Society for thoracic radiology recommended chest imaging for three indications during the pandemic in the following situations [17]: medical triage of patients with a high pre-test probability of COVID-19 in resource-constrained environments; suspected cases in patients at risk of COVID-19 progression; and RT-PCR-confirmed cases of COVID-19 with worsening respiratory status. The French college of radiology also recommended that a chest CT scan should be performed in the event of any therapeutic emergency requiring hospitalisation and/or surgery that cannot wait for the results of a SARS-CoV-2 PCR [24]. A retrospective study found that sensitivity and specificity of LDCT for the diagnosis of COVID-19, respectively, ranged from 75% to 88% in patients admitted to an ED [25]. The accuracy of ULDCT for the diagnosis of COVID-19 pneumonia has also been reported with excellent sensitivity, specificity, positive predictive value, and negative predictive value (86.7%, 93.6%, 91.1%, 90.3%, and 90.2%, respectively) [26]. In this study, reference values for ULDCT were set to 100 kVp and 20 mAs with a pitch of 1.2 and 0.5-s gantry rotation time, and images were reconstructed using sinogram-affirmed iterative reconstruction with a field of view of 450 mm and a matrix size of 512 × 512 pixels [26]. 

A diagnosis score for unenhanced chest CT, named “COVID-19 Reporting and Data System” (CO-RADS), has been developed by the Dutch Radiological society, but it remains to be evaluated in LDCT [27]. This score assessed the suspicion of COVID-19 pneumonia on a scale of 1 (very low) to 5 (very high). CO-RADS was able to distinguish between patients with positive PCR results from those with negative PCR results with an average AUC of 0.91 (95% CI: 0.85, 0.97) [27]. A recent study [28] confirmed that this score is a useful tool allowing radiographers to recognise the classic appearance of COVID-19 on CT in a way comparable to expert radiologists. 

### 2.3. Assessing the Severity of COVID-19 Pneumonia with LDCT

In our experience, thanks to close collaboration between infectologists and radiologists, we performed chest LDCTs on a large cohort of COVID-19 patients [22] and developed a CT score [23]. Our score was designed to measure the anatomical extent of lung impairment with LDCT in patients with a positive SARS-CoV-2 diagnosis. We included eighty patients with positive RT-PCR [23]. We visually classified each lung segment according to the presence of typical features of COVID-19 pneumonia, on a chest LDCT that was performed between Day 3 and Day 11 after the onset of symptoms. A normal chest LDCT was equivalent to 0. Minimal involvement was defined as the presence of a maximum of 10 secondary lobules of any features and was equivalent to 1 (Figure 1a). Intermediate involvement was defined as less than 50% involvement of the segment by any features and was equivalent to 4 (Figure 1b). Severe involvement was defined as more than 50% involvement of the segment by any features and was equivalent to 10 (Figure 1c). The total score was obtained by adding the score of all lung segments. We revealed a positive correlation between the LDCT score and the National Early Warning Score (NEWS), which is predictive for ICU admission (r = 0.48, *p* < 0.001) [23,29]. The strength of our study was reflected in the simplicity and rapidity of our score (10–15 min per patient) and the inclusion of all consecutive patients presenting themselves at our centre with a diagnosis of COVID-19. Our results are consistent with the quantitative CT scores named CT-SS, which is based on the analysis of 20 lung regions with standard dose CT [30]. Retrospectively, the CT-SS at admission was significantly higher for patients requiring high level oxygen therapy than for patients requiring standard care [30]. LDCT have also been used to assess the severity of COVID-19 infected patients by using an automated quantification of CT scans [31].

The extent of lung involvement estimated by CT in COVID-19 pneumonia is now considered as a predictive factor for intubation, prolonged hospital stay, and death [32].

### 2.4. Monitoring COVID-19 Pneumonia with LDCT

During the COVID-19 pandemic, the use of chest CTs has enabled the detection of early fibrotic abnormalities of the lung in several infected patients [33]. This acute fibrotic pattern raised the hypothesis of a potential post-infectious chronic interstitial lung disease, as observed when monitoring patients after MERS and SARS infections [33]. Putative risk factors for developing lung fibrosis after COVID-19 pneumonia have recently been reviewed [34] and suggest the following: older age, smoking, chronic alcoholism, severity of the illness, and length of time on mechanical ventilation [34]. 

Usually, the keystone of evaluating interstitial lung disease is the high-resolution CT (HRCT) and the typical findings are reticulations, traction bronchiolectasis, architectural distortion, and honeycombing. In our centre, we observed early distortive abnormalities in a patient at Weeks 3–4 after the onset of symptoms using LDCT (Figure 2). 

HRCT has been proposed by some authors at six months and one year after recovery from COVID-19 infection [33]. We therefore believe that LDCT could be an interesting tool for monitoring lung abnormalities after a COVID-19 infection in order to reduce the ionising radiation dose. 

### 2.5. Limitations of LDCT for the Diagnosis of Pneumonia

Obesity (BMI > 25) may be one limitation upon dose reduction due to the attenuation of X-rays by thoracic fat [35]. This is important in the context of the COVID-19 pandemic since obesity is a risk factor for severe pneumonia. The other main limitation of LDCT is its inferiority to standard-dose CT angiography for the diagnosis of pulmonary embolisms [36], a frequent and life-threatening complication of COVID-19 infections. Reduced-dose CT angiography leads to significant reductions in diagnostic certainty and image quality [36], and HRCT remains the gold standard in cases where a pulmonary embolism is suspected.

Although chest LDCT may help in the diagnosis of COVID-19 infection, a normal chest CT does not eliminate the diagnosis and can occur in asymptomatic patients or in the first days after the onset of the symptoms [37]. The COVID-19 pneumonia CT pattern is not specific [38], which may lead to false positive results, especially when the prevalence of the virus diminishes in the community. Many acute or chronic, infectious and non-infectious diseases may lead to the same CT findings as COVID-19 infections [38]. For example, ground glass opacities are a common CT finding of pneumocystis and influenza pneumonia. 

## 3. LUS in Treating COVID-19 Pneumonia

### 3.1. Use of Lung US before the COVID-19 Epidemic

Before the COVID-19 pandemic, LUS was already performed in the emergency department (ED) and in intensive care units (ICU) on patients with acute respiratory failure, as a part of the point-of-care ultrasounds (POC US) [39], due to the main qualities of ultrasounds: its immediate availability at any time, the rapidity of the information given, and the lack of need for patient transport in the context of a communicable disease. In recent years, LUS has been shown to be accurate in diagnosing community acquired pneumonias [7]. It is an easy technique with several standardised scanning protocol dividing the thorax into 4–14 zones [40]. 

### 3.2. Using LUS to Diagnose COVID-19 Pneumonia

During the COVID-19 pandemic, LUS has been a useful technique for the early diagnosis of COVID-19 pneumonia [41,42]. Typical findings of LUS consist in an interstitial syndrome, from focal pleural line irregularities to diffuse and confluent B lines, as well as an alveolar syndrome, from small subpleural hypoechoic images to large alveolar consolidations with air bronchograms [43] (Figure 3). As for CT, LUS findings for COVID-19 pneumonia are not specific and have been previously described for interstitial pneumonia caused by chlamydia, pneumocystis, measles, influenza virus A(H7N9) and influenza virus A(H1N1) [40,44,45]. The presence of excessive B lines observed by LUS may also result from non-infectious diseases such as cardiogenic or non-cardiogenic pulmonary oedema, pulmonary fibrosis, flogistic or granulomatous lung interstitial diseases, atelectasis, lymphangitis, and lung contusion [46].

LUS allows a rapid diagnosis of COVID-19 pneumonia and a triage of patients within only 2–3 min in the ED with excellent sensitivity and negative predictive value in comparison to chest CT (93.3% and 94.1%, respectively) [47]. Furthermore, it is a radiation-free technique that may be a valuable alternative to CT scan for the diagnosis of pneumonia in pregnant women and paediatric settings [40]. LUS may also help to identify false-negative results occurring with RT-PCR at the early phase of COVID-19 disease [48]. 

Interestingly, LUS can also predict clinical course and outcomes in COVID-19-positive patients. In a prospective study, the baseline LUS score strongly correlated with the eventual need for invasive mechanical ventilation and death from COVID-19 infection [49]. In one retrospective study, the severity of COVID-19 pneumonia assessed by LUS was highly associated with severity as assessed by chest CT scan, in PCR-positive patients with acute dyspnoea [50]. The LUS score was also associated with the severity of hypoxaemia, and the need for ICU admission and mechanical ventilation [50]. LUS can be performed at the bedside and can diagnose frequent and fatal complications of COVID-19 infection, by visualising pulmonary embolisms or deep venous thrombosis using colour flow Doppler [51]. As for any aetiology of acute respiratory failure, ultrasound can incorporate examinations of both the lungs and the cardiovascular system, to detect myocarditis for example and to improve patient care in the case of COVID-19 infection. Finally, during hospitalisation, LUS can be performed on a daily basis to monitor the extent of COVID-19 pneumonia, as progression assessed by the LUS score may predict the final outcome of the disease in patient with ARDS [52]. 

## 4. Diagnostic Value of LUS and LDCT for Asymptomatic and Mild COVID-19 Infections

Asymptomatic carriers represent 17.9–33.3% of patients with COVID-19 [53,54] and may contribute to the spread of the virus. To our knowledge, the diagnostic and prognostic impact of LDCT for asymptomatic COVID-19 carriers have been rarely reported, with the exception of our study [23] and one prospective preoperative study [55]. The yield of screening for COVID-19 with LDCT in addition to RT-PCR in asymptomatic patients before surgery was low (1.5% patients detected compared to 1.1% patients detected with RT-PCR alone) and did not have any clinical impact for the patients only detected by adding CT [55].

Asymptomatic patients with positive COVID-19 RT-PCR showed pathological LUS findings in 22% of cases, including numerous B lines and pulmonary consolidation in a retrospective study [56]. In comparison to standard dose CT, LUS showed a sensitivity of 66.67% and a positive predictive value of 100% [56]. In a study including pregnant women, the utility of LUS added to the exposure history for predicting a positive COVID-19 RT-PCR was demonstrated for symptomatic women, but not for asymptomatic ones [57].

Finally, LUS and standard dose or LDCT are not routinely recommended as screening tests for COVID-19 or for the diagnosis of asymptomatic careers [58]. 

## 5. Perspectives

Before the SARS-CoV-2 pandemic, several studies suggested the added value of chest ULD and LDCT over CXR for the diagnosis of pneumonia [14]. The COVID-19 pandemic has demonstrated on a large scale the feasibility of using low-radiating chest imaging, and 2020 may have heralded the death of CXR, at least for viral respiratory disease outbreaks. Concerning COVID-19 patients in the ICU, experts from the Fleishner Society made a statement against daily monitoring by CXR [17], which is now considered as obsolete, at least for stable, intubated patients. 

Although LUS is not recommended as the imaging technique of choice for screening and diagnosing COVID-19 pneumonia, it has demonstrated its ability to assess the anatomical severity of pulmonary lesions [50] in the ED and the ICU in order to help in medical triage. 

This is consistent with autopsy results in 28 patients with fatal COVID-19 infection after ARDS showing that the severity of post-mortem LUS findings was correlated with the proportion of diffuse alveolar damage on lung histopathological analysis [59].

The main advantage of this technique is the integration of pulmonary, pleural, cardiac, and vascular impairments in the same examination and its availability at the patient’s bedside. 

We need to improve our knowledge about the value of chest imaging for monitoring the COVID-19 disease. In our opinion, LUS is a promising tool to diagnose and monitor COVID-19 pneumonia, as it is easy to perform by emergency and intensive care doctors or pulmonologists. We also believe that this technique deserves to be used by general practitioners, obstetricians, paediatricians, and nephrologists in haemodialysis centres [40]. Another advantage of thoracic ultrasounds is the possibility of studying the diaphragmatic function, as diaphragmatic impairment has been associated with pneumonia [60] and mechanical ventilation. A recent study has shown that pulmonary diffusion impairment was observed at six months follow-up in 22–56% of patients depending on the initial severity of the disease [61]. In addition, chest HRCT scores using artificial intelligence software found significant abnormalities at six months in these patients [61], but LDCT should be evaluated in this setting. Chest magnetic resonance imaging (MRI) is another non-radiating technique that has shown promising results, especially for chest imaging in children and pregnant women. Despite its poor availability in comparison to LUS or CT, image quality provided by ultra-short echo time (UTE) MRI appears to be equivalent to CT in positive COVID-19 RT-PCR patients [62]. 

Finally, COVID-19 is currently accelerating the transition to low-dose and “no-dose” imaging techniques to explore infectious pneumonia and their long-term consequences. 

## Figures and Tables

**Figure 1 jcm-10-02196-f001:**
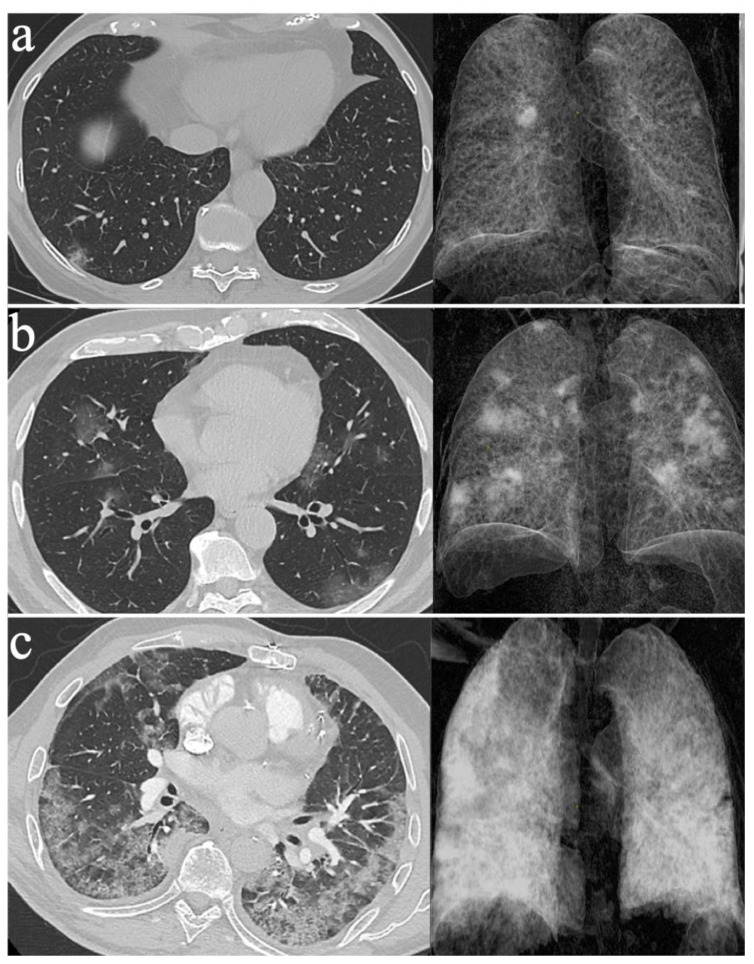
Low-dose non-contrast chest CT scans with 3D volumetric reconstruction in patients with proven COVID-19 infection: (**a**) minimal lung involvement; (**b**) moderate lung involvement; and (**c**) severe lung involvement.

**Figure 2 jcm-10-02196-f002:**
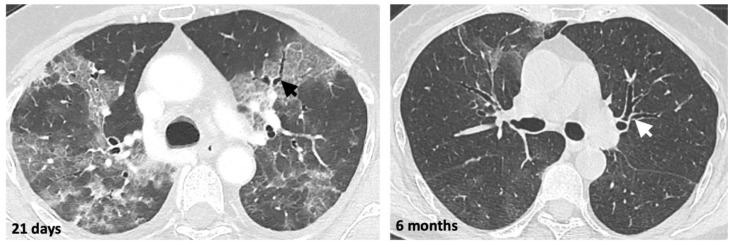
Low-dose non-contrast CT scan of the chest at Day 21 and then six months after the first SARS-CoV-2 PCR-positive result, in a 69-year-old patient. The black and white arrows indicate bronchiectasis.

**Figure 3 jcm-10-02196-f003:**
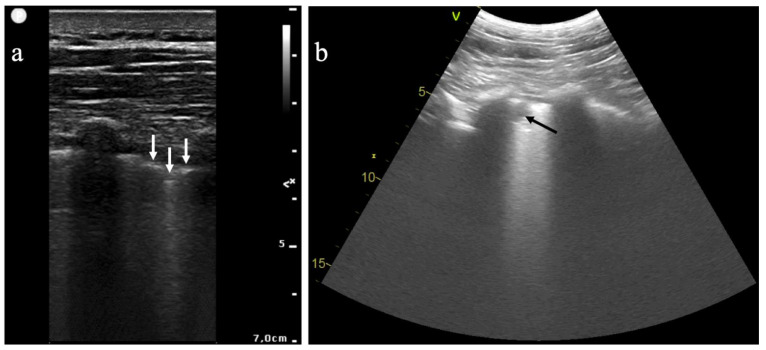
Lung ultrasound images in patients with proven COVID-19 infection: (**a**) longitudinal scan with a high-frequency linear probe, where the white arrows indicate pleural line irregularities; and (**b**) longitudinal scan with a low-frequency convex probe, where the dark arrow indicates a subpleural consolidation.

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
