# Peer review of "Low Dose Chest CT and Lung Ultrasound for the Diagnosis and Management of COVID-19"

_jcm, 2021, doi:10.3390/jcm10102196_

Round 1

Reviewer 1 Report

The present work aims to evaluate the role of LDCT and LUS regarding the diagnosis and management of CoVID-19 pneumonia. The work is a review work. The work is simply structured and linguistically easy to read. 
In my view, the paper has the following weaknesses: In particular, the title is somewhat misleading. Under this title, various other methods, especially the conv. chest X-ray would be expected, since these are much more important than the mentioned LUS and accompany us daily.  MRI also plays an important role in the management of late effects, especially in young patients. Regarding the description of LDCT, it is noticeable that the authors refer very strongly to their own publications and repeat their results here. This can certainly be done to a certain extent, but other results should also be presented in a well-balanced way. For example, the authors' own scoring is promoted, but e.g. CO-RADS is completely missing.

The following minor points in addition: 
- A clear definition regarding LDCT and ULD would be helpful. 
- The mentioned 5.5 mSv are clearly too high and hardly to be found today.

Author Response

  1. The present work aims to evaluate the role of LDCT and LUS regarding the diagnosis and management of CoVID-19 pneumonia. The work is a review work. The work is simply structured and linguistically easy to read. 
    In my view, the paper has the following weaknesses: In particular, the title is somewhat misleading. Under this title, various other methods, especially the conv. chest X-ray would be expected, since these are much more important than the mentioned LUS and accompany us daily.  MRI also plays an important role in the management of late effects, especially in young patients.

Author’s answer: As suggested by reviewer 1, because the title was misleading, we changed it to better describe the topic of the review, as follows: “Low Dose Chest CT and Lung Ultrasound for diagnosis and management of COVID-19 ”

Regarding the other comments: chest X ray lacks sensitivity for the diagnosis of COVID-19 (as for other viral interstitial pneumonia), and this had promoted the development of low dose CT during the pandemic. We added it on page 1 line 40 CXR is the most frequently used chest imaging technique but lacks sensitivity for the diagnosis of COVID-19 pneumonia [3]. » Moreover, as mentioned by reviewer 1, CXR is a well-known tool in daily practice, so that the interest of a new review about appeared less useful to us. Also, our review focuses mainly on the acute disease caused by COVID-19, so that we didn’t chose to include MRI, which is a tool difficult to use in an emergency setting and is restricted to a little number of centers across the world.

We added a sentence about chest MRI in the discussion section Page 9 line 240: “Chest magnetic resonance imaging (MRI) is another non-radiating technique that have also shown promising results especially for chest imaging in children and pregnant women. Despite its poor availability in comparison to LUS or CT, image quality provided by ultra-short echo time (UTE) MRI appears to be equivalent to CT in positive COVID-19 RT-PCR patients[62].”

  1. Regarding the description of LDCT, it is noticeable that the authors refer very strongly to their own publications and repeat their results here. This can certainly be done to a certain extent, but other results should also be presented in a well-balanced way. For example, the authors' own scoring is promoted, but e.g. CO-RADS is completely missing.

Author’s answer: We thanks Reviewer 1 for this comment allowing us to improve the quality and completeness of our MS.

Page 3 line 97 we added a section about other recent studies showing the performances of LDCT for the diagnosis of COVID-19 pneumonia, and we added 2 references:

Page 3 line 97 :

A retrospective study found that sensitivity and specificity of LDCT for the diagnosis of COVID-19 respectively ranged from 75% to 88% in patients admitted to an ED [25]. The accuracy of ULDCT for the diagnosis of COVID-19 pneumonia has also been reported with excellent sensitivity, specificity, positive predictive value, negative predictive value (86.7%, 93.6%, 91.1%, 90.3%, and 90.2% respectively) [26]. In this study, reference values for ULDCT were set to 100 kVp and 20 mAs with a pitch of 1.2 and 0.5-second gantry rotation time and images were reconstructed using sinogram-affirmed iterative reconstruction with a field of view of 450 mm and a matrix size of 512 × 512 pixels [26].  

  1. Desmet J, Biebaû C, De Wever W, et al. Performance of Low-Dose Chest CT as a Triage Tool for Suspected COVID-19 Patients. J Belg Soc Radiol 105. Available at: https://www.ncbi.nlm.nih.gov/pmc/articles/PMC7894373/. Accessed 11 May 2021.

  1. Dangis A, Gieraerts C, De Bruecker Y, et al. Accuracy and Reproducibility of Low-Dose Submillisievert Chest CT for the Diagnosis of COVID-19. Radiol Cardiothorac Imaging 2020; 2:e200196.

Regarding the CO-RADS, we added a sentence to mention it, but with the limitation that it was initially not developed for LDCT

Page  3 we also added the following sentences line 104 about CO-RADS, with two references :

“A diagnosis score for unenhanced chest CT, named “ COVID-19 Reporting and Data System” (CO-RADS) has been developed by the Dutch Radiological society, but remains to be evaluated in LDCT [27]. This score assessed the suspicion of COVID-19 pneumonia on a scale of 1 (very low), to 5 (very high). CO-RADS was able to distinguish between patients with positive PCR results from those with negative PCR results with an average AUC of 0.91 (95% CI: 0.85, 0.97)[27]. A recent study study [28] confirmed that this score is a useful tool allowing radiographers to recognize the classic appearance of COVID-19 on CT in a way comparable to expert radiologists.

  1. Prokop M, van Everdingen W, van Rees Vellinga T, et al. CO-RADS: A Categorical CT Assessment Scheme for Patients Suspected ofHaving COVID-19—Definition and Evaluation. Radiology 2020; 296:E97–E104.
  2. Vicini S, Panvini N, Bellini D, et al. Radiographers and COVID-19 pneumonia: Diagnostic performance using CO-RADS. Radiogr Lond Engl 1995 2021;

We also deleted sentences that referred to our own previous publication in detail, to equilibrate the paragraph :

Page 3 line 153 : “we visually classified each lung segment according to the presence of typical features of COVID-19 pneumonia”.

Page 3 line 159 : “A severe radiological case of COVID-19 pneumonia was defined by a CT score >50/200, which was equivalent to a functional lobectomy. We hypothesized that the extent of pneumonia would be predictive of clinical events.”

Page 3 line 161 : “We also found that dyspnoea, high respiratory rate, hypertension and diabetes were associated with a score > 50. This was consistent with our previous cohort study on patients followed at the IHU Méditerranée Infection, where a normal LDCT was significantly associated with a good clinical outcome, and a CT scan with severe or intermediate lesions was significantly associated with a poor clinical outcome (22.5% vs 1.5% and 37.8% vs 9.3% respectively p<0.005”.

Page 3 line 159 :  The sentence “The total score was obtained by adding the score of all segments for the right and left lungs, with the result ranking between 0 and 200.” was replaced by the following sentence :“The total score was obtained by adding the score of all lung segments.”

  1. The following minor points in addition: 

- A clear definition regarding LDCT and ULD would be helpful. 

Author’s answer: We thank the reviewer for highlighting our lack of precision about this definition. However, we couldn’t find any more specific cutoff for the radiation dose of LDCT and ULD in the literature, and your comment allowed us to add this point of discussion in this section. We also added the effective radiation dose per LDCT reported by the National Lung Screening Trial.

Page 1 line 67 we added the following sentences:

“The definition of low and ultra-low dose chest CT does not include precise quantitative value or technical protocol. It is recognised that the radiation dose of chest LDCT should be half that of standard dose CT [13]. For lung cancer screening, the effective dose estimates were 1.6 mSv 2.4 mSv for one CT for men and women, respectively [6]. Chest ULDCT has a radiation dose equivalent to or lower than CXR (<1mSv) [14]. In comparison, the effective dose for a conventional chest CT is estimated to be 3 to 4.8 mSv [15]. »

  1. The National Lung Screening Trial Research Team. Reduced Lung-Cancer Mortality with Low-Dose Computed Tomographic Screening. N Engl J Med 2011; 365:395–409.

- The mentioned 5.5 mSv are clearly too high and hardly to be found today.

Author’s answer: According to the literature, we were able to find slightly lower radiation dose for conventional chest CT and we modified the sentence page 2 line 71 :

“In comparison, the effective dose for a conventional chest CT is estimated to be 3 to 4.8 mSv [15]. »

Svahn TM, Sjöberg T, Ast JC. Dose estimation of ultra-low-dose chest CT to different sized adult patients. Eur Radiol 2019; 29:4315–4323.

Reviewer 2 Report

In recent years, the use of lung ultrasonography (LUS) has received growing attention in clinical research.
The authors present a review on the role of chest CT scan and LUS in diagnosing infectious lung diseases. The fact that they try to link such an approach to the current Covid-19 pandemic is commendable, and indeed would be  desirable for societal reasons.

The topic is of considerable clinical interest. The writing is clear and easily understandable. I found the article interesting and thoroughly enjoyed reading the manuscript.

Specific comments:
- There are some minor grammatical errors or typos. The writing may benefit a review by a professional writer.
- Chest X-rays cannot provide enough information at any time and did not show an high sensitivity in detecting COVID-19. However, Authors should include a couple of sentences and references about covid-19, and specifically about the inferiority of chest x-ray in detecting covid-19 pneumonia. An example:
Gibbons RC, Magee M, Goett H, Murrett J, Genninger J, Mendez K, Tripod M, Tyner N, Costantino TG. Lung Ultrasound vs. Chest X-Ray Study for the Radiographic Diagnosis of COVID-19 Pneumonia in a High-Prevalence Population. J Emerg Med. 2021 Feb 4:S0736-4679(21)00101-3.
- LUS may help in early diagnosis, therapeutic decisions and follow-up monitoring of COVID-19 pneumonia not only in the critical care setting, but also in several other different settings (general practitioners’ offices, nursing homes, emergency departments, general internal medicine wards, pulmonology wards, hemodialysis units, obstetrics and paediatrics). Authors should include this concept and the following reference:
Allinovi M, Parise A, Giacalone M, Amerio A, Delsante M, Odone A, Franci A, Gigliotti F, Amadasi S, Delmonte D, Parri N, Mangia A. Lung ultrasound may support diagnosis and monitoring of COVID-19 pneumonia. Ultrasound Med Biol 2020 Jul 20:S0301-5629(20)30333-1.
- Discussion, page 6, line 166. The following sentence is not completely correct “As for CT, LUS findings for COVID-19 pneumonia are not specific and have been previously described during influenza pandemics”. In fact, a LUS pattern similar to COVID-19 was previously described for interstitial pneumonia caused by Chlamydia, Pneumocystis, measles, influenza virus A H7N9 and influenza virus H1N1. Even this concept should be included:
Allinovi et al, Ultrasound Med Biol 2020
Lo Giudice V, Bruni A, Corcioni E, Corcioni B. Ultrasound in the evaluation of interstitial pneumonia. J Ultrasound. 2008 Mar; 11(1): 30–38.
- A specific paragraph should be dedicated to positive LUS assessments (B-lines and abnormalities in pleural line) and chest CT scan assessments in asymptomatic or olygosymptomatic patients positive for covid-19. Are they sensitive enough to diagnose or suspect COVID-19 early or in mild forms?
- Discussion, page 6, line 159. The following sentence is not completely correct “It is an easy technique with a standardised scanning protocol for each of the 12 lung quadrants”. In fact, there’s not only the 12-zone technique, but several LUS approaches with different number of zones assessed.

Limitations:
-  This paper is neither the first paper that compares chest CT scan and LUS in detecting viral pneumonia, nor the first in the contest of LUS and COVID-19.
-  The specificity of B-lines at LUS is low: in addition to pulmonary congestion, these are visible in pulmonary fibrosis, flogistic or granulomatous lung interstitial diseases, atelectasis, lymphangitis, lung contusion, cardiac failure and acute respiratory distress syndrome. Authors should include this sentence into the text, and the following reference:
Dietrich CF, Mathis G, Blaivas M, Volpicelli G, Seibel A, Wastl D, Atkinson NS, Cui XW, Fan M, Yi D. Lung B-line artefacts and their use. J Thorac Dis. 2016 Jun;8(6):1356-65.

Author Response

In recent years, the use of lung ultrasonography (LUS) has received growing attention in clinical research.
The authors present a review on the role of chest CT scan and LUS in diagnosing infectious lung diseases. The fact that they try to link such an approach to the current Covid-19 pandemic is commendable, and indeed would be desirable for societal reasons.
The topic is of considerable clinical interest. The writing is clear and easily understandable. I found the article interesting and thoroughly enjoyed reading the manuscript.

Specific comments:

- There are some minor grammatical errors or typos. The writing may benefit a review by a professional writer.

Author’s answer: The MS has been reviewed and corrected by a native English speaker

- Chest X-rays cannot provide enough information at any time and did not show a high sensitivity in detecting COVID-19. However, Authors should include a couple of sentences and references about covid-19, and specifically about the inferiority of chest x-ray in detecting covid-19 pneumonia. An example:
Gibbons RC, Magee M, Goett H, Murrett J, Genninger J, Mendez K, Tripod M, Tyner N, Costantino TG. Lung Ultrasound vs. Chest X-Ray Study for the Radiographic Diagnosis of COVID-19 Pneumonia in a High-Prevalence Population. J Emerg Med. 2021 Feb 4:S0736-4679(21)00101-3.

Author’s answer: We thank reviewer 2 for his suggestions, which will improve the quality of the MS.  We added it as follows:  Page 1 line 38:

“CXR is the most frequently used chest imaging technique but lacks sensitivity for the diagnosis of COVID-19 pneumonia [3] Gibbons RC, Magee M, Goett H, et al. Lung Ultrasound vs. Chest X-Ray Study for the Radiographic Diagnosis of COVID-19 Pneumonia in a High-Prevalence Population. J Emerg Med 2021;”

- LUS may help in early diagnosis, therapeutic decisions and follow-up monitoring of COVID-19 pneumonia not only in the critical care setting, but also in several other different settings (general practitioners’ offices, nursing homes, emergency departments, general internal medicine wards, pulmonology wards, hemodialysis units, obstetrics and paediatrics). Authors should include this concept and the following reference:
Allinovi M, Parise A, Giacalone M, Amerio A, Delsante M, Odone A, Franci A, Gigliotti F, Amadasi S, Delmonte D, Parri N, Mangia A. Lung ultrasound may support diagnosis and monitoring of COVID-19 pneumonia. Ultrasound Med Biol 2020 Jul 20:S0301-5629(20)30333-1.

Author’s answer: Page 7 line 187 (Paragraph 3.2 Using LUS to diagnose COVID-19 pneumonia) the following sentence is added with the suggested reference :

“Furthermore, it is a radiation-free technique that may be a valuable alternative to CT scan for the diagnosis of pneumonia in pregnant women and pediatric settings [39] Allinovi M, Parise A, Giacalone M, et al. Lung Ultrasound May Support Diagnosis and Monitoring of COVID-19 Pneumonia. Ultrasound Med Biol 2020; 46:2908–2917”

Page 9 line 232 (Section 4. Perspectives) the following sentence was added with the same reference :

“In our opinion, LUS is a promising tool to diagnose and monitor COVID-19 pneumonia, as it is easy to perform by emergency and intensive care doctors or pulmonologists. We also believe that this technique deserves to be used by general practitioners, obstetricians, pediatricians and nephrologists in hemodialysis centers [39].”

- Discussion, page 6, line 166. The following sentence is not completely correct “As for CT, LUS findings for COVID-19 pneumonia are not specific and have been previously described during influenza pandemics”. In fact, a LUS pattern similar to COVID-19 was previously described for interstitial pneumonia caused by Chlamydia, Pneumocystis, measles, influenza virus A H7N9 and influenza virus H1N1. Even this concept should be included:
Allinovi et al, Ultrasound Med Biol 2020
Lo Giudice V, Bruni A, Corcioni E, Corcioni B. Ultrasound in the evaluation of interstitial pneumonia. J Ultrasound. 2008 Mar; 11(1): 30–38.

Author’s answer:  Page 7 line 179 (Paragraph 3.2 Using LUS to diagnose COVID-19 pneumonia)

 the following sentence was modified as follows: “As for CT, LUS findings for COVID-19 pneumonia are not specific and have been previously described for interstitial pneumonia caused by Chlamydia, Pneumocystis, measles, influenza virus A(H7N9) and influenza virus A(H1N1) [37,38] »

Lo Giudice V, Bruni A, Corcioni E, Corcioni B. Ultrasound in the evaluation of interstitial pneumonia. J Ultrasound 2008; 11:30–38. Allinovi M, Parise A, Giacalone M, et al. Lung Ultrasound May Support Diagnosis and Monitoring of COVID-19 Pneumonia. Ultrasound Med Biol 2020; 46:2908–2917.

The following reference was deleted (redundacy):

Testa A, Soldati G, Copetti R, Giannuzzi R, Portale G, Gentiloni-Silveri N. Early recognition of the 2009 pandemic influenza A (H1N1) pneumonia by chest ultrasound. Crit Care 2012;16:R30.

- A specific paragraph should be dedicated to positive LUS assessments (B-lines and abnormalities in pleural line) and chest CT scan assessments in asymptomatic or olygosymptomatic patients positive for covid-19. Are they sensitive enough to diagnose or suspect COVID-19 early or in mild forms?

  1. Author’s answer: The suggested paragraph, named “Diagnostic value of LUS and LDCT for asymptomatic and mild COVID-19 infections”has been added page 8 line 205

“Asymptomatic carriers represent 17.9 to 33.3% of patients with COVID-19 [53,54] and may contribute to the spread of the virus. To our knowledge, the diagnostic and prognostic impact of LDCT for asymptomatic COVID-19 carriers have been rarely reported, with the exception of our study [23] and one prospective preoperative study [55]. The yield of screening for COVID-19 with LDCT in addition to RT-PCR in asymptomatic patients before surgery was low (1.5% patients detected compared to 1.1% patients detected with RT-PCR alone) and did not have any clinical impact for the patients only detected by adding CT [55].

Asymptomatic patients with positive COVID-19 RT-PCR showed pathological LUS findings in 22% of cases, including numerous B lines and pulmonary consolidation in a retrospective study [56]. In comparison to standard dose CT, LUS showed a sensitivity of 66.67% and a positive predictive value of 100% [56]. In a study including pregnant women, the utility of LUS added to the exposure history for predicting a positive COVID-19 RT-PCR was demonstrated for symptomatic women, but not for asymptomatic ones [57].

Finally, LUS and standard dose or LDCT are not routinely recommended as screening tests for COVID-19 or for the diagnosis of asymptomatic careers [58]. “

- Discussion, page 6, line 159. The following sentence is not completely correct “It is an easy technique with a standardised scanning protocol for each of the 12 lung quadrants”. In fact, there’s not only the 12-zone technique, but several LUS approaches with different number of zones assessed.

Author’s answer: Page
7 line 173 (Paragraph 3.1Use of lung US before the COVID-19 epidemic) the following sentence and citation have been modified as suggested :

It is an easy technique with several standardised scanning protocol dividing the thorax in 4 to 14 zones [33]

Allinovi M, Parise A, Giacalone M, et al. Lung Ultrasound May Support Diagnosis and Monitoring of COVID-19 Pneumonia. Ultrasound Med Biol 2020; 46:2908–2917.

Limitations:
- This paper is neither the first paper that compares chest CT scan and LUS in detecting viral pneumonia, nor the first in the contest of LUS and COVID-19.

-  The specificity of B-lines at LUS is low: in addition to pulmonary congestion, these are visible in pulmonary fibrosis, flogistic or granulomatous lung interstitial diseases, atelectasis, lymphangitis, lung contusion, cardiac failure and acute respiratory distress syndrome. Authors should include this sentence into the text, and the following reference:
Dietrich CF, Mathis G, Blaivas M, Volpicelli G, Seibel A, Wastl D, Atkinson NS, Cui XW, Fan M, Yi D. Lung B-line artefacts and their use. J Thorac Dis. 2016 Jun;8(6):1356-65. 

Author’s answer:

Page 7 line 181 the sentence suggested has been added :

“The presence of excessive B lines observed by LUS may also results from non-infectious diseases such as cardiogenic or non-cardiogenic pulmonary oedema, pulmonary fibrosis, flogistic or granulomatous lung interstitial diseases, atelectasis, lymphangitis and lung contusion [39].”

Reviewer 3 Report

The manuscript is well written and deals with all the current problems in the diagnosis and management of COVID-19 pneumonia. I would suggest including technical parameters of acquisition for the low-dose and ultra-low-dose CT examinations. 

Author Response

Reviewer 3. The manuscript is well written and deals with all the current problems in the diagnosis and management of COVID-19 pneumonia. I would suggest including technical parameters of acquisition for the low-dose and ultra-low-dose CT examinations. 

Author’s answer: Technical acquisition parameters for LD and ULDCT were added :

Page 3 line 86: “LDCT scans were performed with the following parameters : detector collimation: 0.625 mm; field of view: 500 mm; matrix: 512x512; pitch:1.375; gantry speed 0.35s; 120 KV; 45 mAs; and reconstructed slice thickness 1.2mm. [23].

  1. Leger T, Jacquier A, Barral P-A, et al. Low-dose chest CT for diagnosing and assessing the extent of lung involvement of SARS-CoV-2 pneumonia using a semi quantitative score. PloS One 2020; 15:e0241407.

Page 3 line 101 :

In this study, reference values were set to 100 kVp and 20 mAs with a pitch of 1.2 and 0.5-second gantry rotation time and images were reconstructed using sinogram-affirmed iterative reconstruction with a field of view of 450 mm and a matrix size of 512 × 512 pixels [26].

  1. Dangis A, Gieraerts C, De Bruecker Y, et al. Accuracy and Reproducibility of Low-Dose Submillisievert Chest CT for the Diagnosis of COVID-19. Radiol Cardiothorac Imaging 2020; 2:e200196.

Round 2

Reviewer 1 Report

The paper has been substantially improved by the authors. Most of my comments were implemented by the authors and included in the new version. I think that with the adjustment of the title, the review is more clearly focused and consistent with the content. The general overview of the two modalities is now better designed.